

# On the value of water quality data and informative flow states in karst modelling

Andreas Hartmann[1,2], Juan Antonio Barberá[3], Bartolomé Andreo[3]

[1]Faculty of Environment and Natural Resources, University of Freiburg, Germany
[2]Department of Civil Engineering, University of Bristol, UK
[3]Department of Geology and Centre of Hydrogeology of the University of Malaga (CEHIUMA), Malaga 29071, Spain

*Correspondence to*: Andreas Hartmann (andreas.hartmann@hydrology.uni-freiburg.de)

**Abstract.** If properly applied, karst hydrological models are a valuable tool for karst water resources management. If they are able to reproduce the relevant flow and storage processes of a karst system, they can be used for prediction of water resources availability when climate or land use are expected to change. A common challenge to apply karst simulation models is the limited availability of observations to identify their model parameters. In this study, we quantify the value of information when water quality data ($NO_3^-$ and $SO_4^{-2}$) is used in addition to discharge observations to estimate the

parameters of a process-based karst simulation model at a test site in Southern Spain. We use a three-step procedure to (1) confine an initial sample of 500,000 model parameter sets by discharge and water quality observations, (2) identify alterations of model parameter distributions through the confinement, and (3) quantify the strength of the confinement for the model parameters. We repeat this procedure for flow states, at which the system discharge is controlled by the unsaturated zone, the saturated zone, and the entire time period including times when the spring is influenced by a nearby river. Our

results indicate that $NO_3^-$ provides most information to identify the model parameters controlling soil and epikarst dynamics during the unsaturated flow state. During the saturated flow state, $SO_4^{-2}$ and discharge observations provide the best information to identify the model parameters related to groundwater processes. We found reduced parameter identifiability when the entire time period is used as the river influence disturbs parameter estimation. We finally show that most reliable simulations are obtained when a combination of discharge and water quality date is used for the combined unsaturated and

saturated flow states.

## 1   Introduction

It is estimated that around 10-15% of emerged Earth surface is covered by soluble rocks that are susceptible to be karstified (Ford and Williams, 2013). Today, aquifers developed in such type of rocks roughly supply with groundwater to a quarter of

world´s population. The importance of groundwater resources from karst aquifers is not only limited to satisfy the fresh water demand of large regions with some millions of inhabitants (e.g. Austria or Slovenia), but also it guarantees the water supply in small settlements where karst waters are the only source of drinking water.



The intrinsic characteristics of karst aquifers such as the development of a secondary porosity through enlarged conduits and fractures, as well as the duality in the recharge (diffuse *vs* concentrated), result in a high permeability media (Bakalowicz, 2005; White and White, 2003). The shallower parts of the aquifers, including soil and epikarst (i.e. unsaturated zone), play a key role for the understanding of system functioning. Epikarst is characterized by slow percolation of air and water into narrow fissures, inducing water storage, and by a rapid drainage through connected conduit system promoting flow concentration (Aquilina et al., 2006; Ford and Williams, 2013; Labat et al., 2000). Thus, in the first top meters of aquifer rock, biogeochemical processes occur in a multiphase environment (gas, liquid and solid) and recharge waters rapidly acquire their chemical composition, keeping practically unaltered until reach the emergence points. Rapid drainage impedes that such physical-chemical processes may attenuate naturally a potential contaminant entering into the system. Therefore, karst aquifers are especially vulnerable to the contamination despite that the unsaturated zone, jointly with soil and epikarst, acts chemically as reaction layer able to modify the groundwater quality in a substantial way.

Simulation models are a common tool to address water management questions such as the impacts of climate and land use changes on karst water resources (Hartmann et al., 2014a). In order to provide reliable predictions those models need to include the most relevant processes of karst systems and various approaches have been developed to include karst processes in distributed and lumped karst simulation models (Ghasemizadeh et al., 2012; Hartmann et al., 2014a; Kovacs and Sauter, 2007; Sauter et al., 2006). The choice of the model approach is usually due to the required purpose. A key challenge in all of these karst modelling approaches is the identification of the model parameters. Methods to explore and analyse karst systems can provide prior knowledge on karst system properties (Goldscheider and Drew, 2007) that can be used to gain prior information of karst model parameters such as hydraulic conductivities or catchment boundaries. However, capturing the entire heterogeneity of karst systems with those methods is commonly impossible (Hartmann et al., 2013b) and inverse parameter estimation schemes, for instance automatic calibration by observed discharge, have to be applied.

Work with automatic calibration approaches early showed that using only discharge observations for model calibration allows to identify up six model parameters (Jakeman and Hornberger, 1993; Wheater et al., 1986; Ye et al., 1997). More recent work also revealed that including disinformative periods in the calibration, i.e. periods when errors in the observation can be expected, may significantly bias the results of model calibration and evaluation of hydrological models (Beven et al., 2011; Beven and Westerberg, 2011; Kauffeldt et al., 2013). Due to the complexity of karst processes, karst models usually require more than 6 model parameters to reflect the most important hydrological processes. Some studies tried to compensate for this apparent lack of information by using auxiliary data such as gravimetric information (Mazzilli et al., 2012), artificial tracer experiments (Hartmann et al., 2012; Oehlmann et al., 2015), or hydrochemical information (Charlier et al., 2012; Hartmann et al., 2016, 2013a). However, to our knowledge the problem of disinformative observations, either discharge observations or auxiliary information, has not been addressed explicitly in karst modelling studies.

This study proposes a new approach to quantitatively assess the information content of discharge and hydrochemical information for karst model calibration including periods with disinformative observations. A process-based model is used to simulate the hydrodynamic and hydrochemical ($NO_3^-$ and $SO_4^{-2}$) behaviour of a karst system, at which the unsaturated zone



dynamics dominates under recharge conditions, controlling groundwater flow and solute transport processes. During specific periods, the discharge and chemistry of the system is influenced by the surface flow of a nearby river, which constitutes disinformative periods for model parameter estimation. A new parameter estimation approach is employed to estimate the information content of the different types of calibration data during pre-defined flow states that focus on time periods dominated by unsaturated zone discharge, saturated zone discharge, and periods that include the disinformative observations. Even though applied to one particular study site this approach can easily be transferred to any hydrological system where different observation types are available for calibration.

## 2    Study site description

The experimental area is located in the Eastern Ronda Mountains, at the NW of Málaga province (S Spain). It consists of steep and rugged NE-SW oriented reliefs (e.g. Sierra Blanquilla), reaching a maximum height of 1,428 m a.s.l. (Viento peak; Figure 1). Geologically, three main stratigraphic groups can be differentiated (Cruz-Sanjulián, 1974; Martín-Algarra, 1987, Figure 1): (i) clays and evaporites of upper Triassic age (the older formation); (ii) a thick (up to 500 m) carbonate sequence of Jurassic dolostones and limestones forming the main aquifer (i.e. Sierra Blanquilla); and (iii) Cretaceous-Paleogene marls and marly limestones as the uppermost materials. The geological structure of Sierra Blanquilla is constituted by a NE-SW oriented box-shaped anticline, plunging towards NE (Martín-Algarra, 1987), with a flat and wide hinge, as well as subvertical flanks. The folded structure is also fractured by two set of faults N50-70E and N150E oriented (Fernández, 1980). From the point of view of the karst landscape development, in plateau areas the horizontal bedding planes of carbonate exposures jointly to the high precipitation rate have favoured the formation of exokarstic features including karrenfields, dolines, uvalas, shafts and swallets, as result of intense karstification processes.

## 2.1    Karst hydrogeology

Sierra Blanquilla carbonate aquifer is permeable by fracturation and karstification. Recharge is mostly produced by rainwater infiltration through the carbonate exposures, although seepage from a losing river and streams also account for groundwater input (Barberá and Andreo, 2012, 2015). Natural groundwater discharge is preferentially conducted toward the SE border of the aquifer (Figure 1), through several springs that constitute the discharge area towards the Turón river valley (Barberá, 2014). Among them, El Burgo (BG, 600 m a.s.l.) and Hierbabuena (HB, 645 m a.s.l.) perennial springs drain most of the groundwater of the hydrogeological system (Figure 1). During high flow periods, when the total flow of the BG and HB springs exceeds 1.1 $m^3 \cdot s^{-1}$, two overflow springs (OfsI, 655 m a.s.l.; and OfsII, 670 m a.s.l.), located upstream of the permanent ones, activate after heavy rainfall events (Barberá and Andreo, 2015). Low flow is established when the permanent groundwater flow (from BG and HB springs) is below to 0.2 $m^3 \cdot s^{-1}$.

The main hydrological feature in the test site, Turón River, crosses intermittently the carbonate exposures in the southern border of Blanquilla aquifer (Figure 1). The surface flow has been demonstrated to alter the hydrodynamic functioning of



both perennial springs (Barberá and Andreo, 2015), which are partly affected by the existence of two regulation dams (20-25 m high) built over the Turón riverbed, just several tens of meters downstream from the springs (Figure 1). In high flow periods, both headwaters and groundwater discharge from Sierra Blanquilla aquifer maintain the river flow, while during low flow conditions, the Turón river is exclusively fed by karst groundwater.




**Figure 1: Geographic, geological and hydrogeological features of Sierra Blanquilla carbonate aquifer.**



## 2.2    Dominant hydrogeological processes

Electrical conductivity (EC) has been used as global physical-chemical marker for distinguishing the hydrochemical states that characterize El Burgo spring discharge. Generally, EC peaks are concomitant with maximum spring discharge at event scale, which evidence that more mineralized groundwater is drained immediately after each rainfall episode (green shaded areas in Figure 2). Barberá and Andreo (2015) stated that this high EC groundwater is also characterized by higher Alkalinity and logPCO$_2$ values and higher Ca$^{+2}$ and TOC contents, suggesting predominant limestone dissolution in the shallower parts of the aquifer. This spring behaviour reflect a functioning based on a "piston effect", by which groundwater stored in the epikarst reservoir is pulled out to the unsaturated and saturated zone until the discharge zone by a subsequent recharge pulse. Therefore, unsaturated flow dominates under high water conditions in El Burgo spring (state 1 - unsaturated zone, in Figure 2).

Under low flow conditions (no rainfall, grey shaded areas in Figure 2), EC levels in groundwater remain quite stable in the range of 320-330 µS/cm. This provides the chemical baseline of the system (state 2 - saturated zone, in Figure 2), which is dependent on the accumulated rainfall on each hydrological year. The lower and less variable EC values of groundwater compared with those obtained under high wtaer conditions can be explained by the loss of aggressiveness of groundwater (degassed waters respect to CO$_2$) flowing through the system as consequence of the lack of aquifer recharge (Barberá and Andreo, 2015). Therefore, groundwater drainage under low water conditions consists of a system of slower flows coming from capacitive compartments of the aquifer (matrix). In these circumstances, the functioning of the hydrogeological system is mainly dominated by the saturated zone (state 2 - saturated zone, in Figure 2).

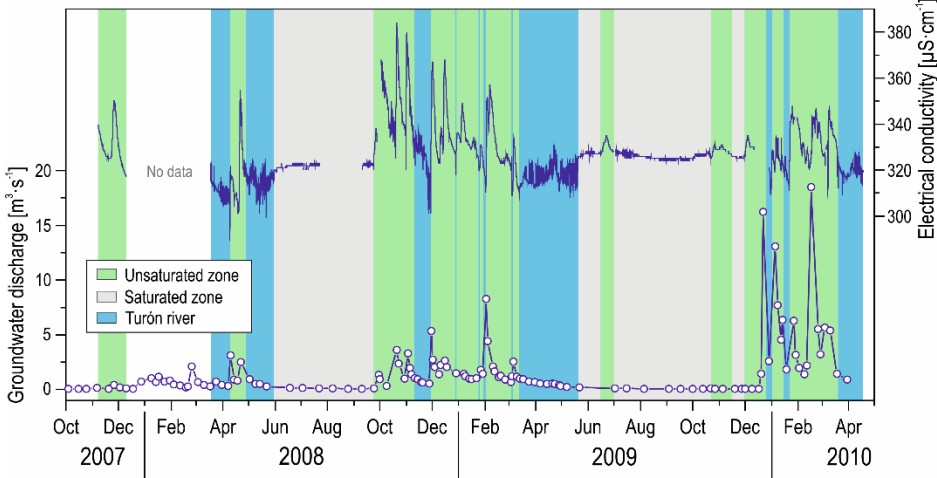

**Figure 2: Decomposition of El Burgo spring flow in selected hydrochemical states from EC and discharge time series: (1) unsaturated zone dominates discharge; (2) saturated zone dominates discharge; and (3) discharge (and EC) influenced by Turón river flow**

Marked dilutions in groundwater mineralization (below the chemical baseline of the system), which very often occur during the spring recession after flood events, are also observed in the chemograph of El Burgo spring (preferentially from March to





June, in Figure 2). Since Turón river waters are less mineralized than groundwater and that the temporary storage of surface water in the nearby river dam favours water mixing, surface water dilutes groundwater from the spring (state 3 – Turón river, in Figure 2). This occurs when the river stage is higher than groundwater level in the discharge zone, promoting water flow towards the aquifer (Barberá and Andreo, 2015).

## 3    Methodology

### 3.1    Available data

Continuous daily measurements of precipitation and air temperature were recorded at Añoreta weather station (Figure 1) and discrete sampling campaigns for meteoric water chemistry ($NO_3^-$ and $SO_4^{-2}$, among other) were performed in a rain collector installed to the north of Viento peak (Figure 1), from August 2007 to April 2010. From meteorological data, potential
evapotranspiration was calculated using Thornthwaite's approach (Thornthwaite, 1948). Discontinuous measurements of Turón river flow in two selected sections ($T_{up}$ and $T_{dn}$; Figure 1), upstream and downstream of the permanent and temporary springs, were conducted during the same study period to quantify the net groundwater discharge from Sierra Blanquilla aquifer. Simultaneously, a representative sampling of the chemical composition ($NO_3^-$ and $SO_4^{-2}$) of karst groundwater was performed (daily to biweekly) at El Burgo spring. Additionally, hourly data of EC was recorded at this outlet. Detailed
methodological procedures can be found in Barberá and Andreo (2015). The environmental tracers $NO_3^-$ and $SO_4^{-2}$ were chosen as complementary time series for the model development because they are expected to provide distinctive chemical signatures that characterize flow and transport processes in the soil and epikarst (nitrogen cycling) and saturated zone (dissolution of evaporites at the aquifer basement) of Sierra Blanquilla aquifer. Table 1 provides a summary of all available data. In addition, the information about the three differentiated flow states of the system (subsection 2.2) was used to provide
an independent consideration of observations that can be attributed to time periods of state 1 (unsaturated zone), state 2 (saturated zone), and all states including the period influenced by Turón River dynamics (state 3).

**Table 1: Main characteristics of the time series of hydrodynamic and hydrochemical data used in this study.**

| Sampling site | Parameter | Unit | n | Max | Min | Mean | CV (%) | Average sampling frequency | Period |
|---|---|---|---|---|---|---|---|---|---|
| Añoreta weather st. | Rainfall (accumulated) | mm·day$^{-1}$ | 959 | 71 | 0 | 3.3 | - | 1 day | 16/08/2007 - 31/03/2010 |
| | Air temperature (daily mean) | ºC | 959 | 14.9 | 2.6 | 8 | - | 1 day | 16/08/2007 - 31/03/2010 |
| Viento rain collector | $NO_3^-$ | mg·l$^{-1}$ | 38 | 23 | 0 | 3 | 2 | 15 days * | 04/10/2007 - 16/02/2010 |
| | $SO_4^{-2}$ | mg·l$^{-1}$ | 38 | 4.9 | 0.3 | 1.2 | 1 | 15 days * | 04/10/2007 - 16/02/2010 |
| Turón river | Discharge (GW component) | m$^3$·s$^{-1}$ | 132 | 18,5 | 0.06 | 1,63 | 169 | 7 days | 16/08/2007 - 30/03/2010 |
| El Burgo spring | Electrical conductivity (EC) | µS·cm$^{-1}$ | 17,296 | 384 | 288 | 326 | 3 | 1 hour | 07/11/2007 - 15/04/2010 |
| | $NO_3^-$ | mg·l$^{-1}$ | 130 | 21.2 | 0.8 | 5.1 | 56 | 8 days | 01/08/2007 - 30/03/2010 |
| | $SO_4^{-2}$ | mg·l$^{-1}$ | 130 | 24.4 | 4.2 | 11.4 | 49 | 8 days | 01/08/2007 - 30/03/2010 |

**(\*) Sampling frequency was dependent on the occurrence of rainfall episodes**





## 3.2    The model

VarKarst model was previously developed at a neighbour karst system in Southern Spain (Hartmann et al., 2013a) and it was successfully applied at different karst systems around Europe (Brenner et al., 2016; Hartmann et al., 2014b, 2013b; Mudarra et al., n.d.). It includes the variability of karst system properties by statistical distribution functions (Figure 3). Explicitly, it considers the spatial variability of (i) soil and epikarst depths, (ii) fractions of concentrated and diffuse recharge to the groundwater, (iii) epikarst hydrodynamics, and (iv) groundwater hydrodynamics by distribution functions that are applied to a set of $N$ model compartments. This allows the simulation of variably dynamic pathways of water and solutes through the karst system. The detailed equations of the model in the appendix and a list of all model parameters including their description are provided in Table 2.

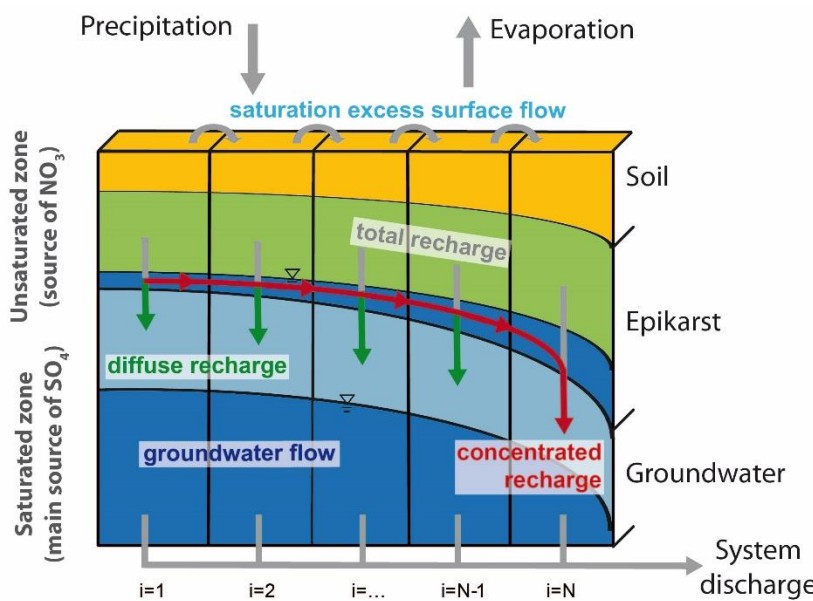

**Figure 3: Schematic representation of the VarKarst model structure (modified from Hartmann et al., 2013a; modified)**

## 3.3    Parameter estimation for the distinctive flow states and different observation types

The low resolution of observed discharge and hydrochemistry, as well as the complex karstic setting of the study site creates a rather uncertain environment for modeling. For that reason, a traditional multi-objective parameter estimation was omitted as in previous studies (Hartmann et al., 2016, 2013a). Instead, a parameter estimation scheme considering "soft rules" was used to confine a large uniformly sampled set of model parameters the fore explicitly allowing for some uncertainty to remain but to be quantified. A similar approach was already applied successfully for cases with similarly complex modelling domains: another karst system in Southern Spain (Mudarra et al., n.d.) and a large-scale karst groundwater recharge study (Hartmann et al., 2015).





As a measure of performance, the Kling-Gupta efficiency KGE (Gupta et al., 2009) is used. It is defined to show numbers approaching 1 for the best simulations:

$$KGE = 1 - \sqrt{(r-1)^2 + (\alpha-1)^2 + (\beta-1)^2} \qquad (1)$$

with $\alpha = \dfrac{\sigma_S}{\sigma_O}$ and $\beta = \dfrac{\mu_S}{\mu_O}$. $\qquad (2)$

$r$ expresses the linear correlation coefficient between simulations and observations, while $\mu_s/\mu_o$ and $\sigma_s/\sigma_o$ are defined as the mean and standard deviation of simulations and observations, respectively. Consequently, $\alpha$ expresses the similarity of simulated and observed variability, while $\beta$ quantifies the bias between them.

**Table 2: Description of model parameters, ranges for parameters estimation and average values found for the combined**
**unsaturated and saturated flow states, and the entire time period including the disinformative period of river influence.**

| Parameter | Description | Unit | Parameter ranges | | Combined unsat. and sat. states | All states |
|---|---|---|---|---|---|---|
| | | | lower | upper | mean* | mean* |
| A | Recharge area | [km²] | 30 | 80 | 55.5 | 57.5 |
| $V_S$ | Mean soil storage capacity | [mm] | 0 | 250 | 159.9 | 174.1 |
| $V_E$ | Mean epikarst storage capacity | [mm] | 0 | 250 | 23.5 | 75.8 |
| $a_{SE}$ | Soil/epikarst depth variability constant | [-] | 0 | 3 | 0.6 | 1.8 |
| $K_E$ | Epikarst mean storage coefficient | [d⁻¹] | 15 | 65 | 49.4 | 43.4 |
| $a_f$ | Recharge separation variability constant | [-] | 0 | 3 | 1.4 | 1.3 |
| $K_C$ | Conduit storage coefficient | [d⁻¹] | 1 | 25 | 5.7 | 12.4 |
| $a_{GW}$ | Groundwater variability constant | [-] | 0 | 3 | 1.8 | 1.3 |
| $c_{SO4}$ | Mean equilibrium concentration of $SO_4^{2-}$ | [mg l⁻¹] | 0 | 100 | 16.6 | 22.0 |
| $a_{SO4}$ | $SO_4^{2-}$ variability constant | [-] | 0 | 3 | 0.6 | 1.4 |
| $KGE_Q$ | performance concerning discharge | [-] | 0 | 1 | 0.37 | 0.36 |
| $KGE_{NO3}$ | performance concerning $NO_3$ | [-] | 0 | 1 | 0.47 | 0.32 |
| $KGE_{SO4}$ | performance concerning $SO_4^{2-}$ | [-] | 0 | 1 | 0.58 | 0.40 |

\* variability of model parameters shown in Figure 5

      For parameter estimation, an initial sample of 250,000 parameter sets was created from predefined ranges (Table 2) that were chosen by prior knowledge and previous model experiences in the same region (Hartmann et al., 2014b, 2013a; Mudarra et al., n.d.). A 4-year warm up period was set up and the model was run 250,000 times with the initial parameter
sample. Using the observed time series, the Kling Gupta Efficiency was calculated for each of the simulation runs: $KGE_Q$ (groundwater discharge), $KGE_{NO3}$ ($NO_3^-$ concentrations) and $KGE_{SO4}$ ($SO_4^{-2}$ concentrations). Similar to Choi and Beven (2007) "soft rules" were used to reduce the initial sample of parameters in four steps:

1. All parameters sets from the initial sample with $KGE_Q < 0.2$ were discarded



2.  All parameters sets from the initial sample with $KGE_{NO3} < 0.2$ were discarded

3.  All parameters sets from the initial sample with $KGE_{SO4} < 0.2$ were discarded

4.  All parameters sets from the initial sample with $KGE_Q$, $KGE_{NO3}$, and $KGE_{SO4} < 0.2$ at the same time were discarded

The threshold value of 0.2 was found by preliminary analysis. The same procedure is repeated four times for observations falling into the unsaturated flow state, the saturated flow state, the combined unsaturated and saturated flow state and into the entire time period including the hydrodynamic state defined by influence of Turón river flow on groundwater discharge. For each of these time periods the four soft rules will result in a reduction of the initial sample and the prior ranges of the model parameters will experience a confinement (Hartmann et al., 2015).

## 3.4 Evaluation of information content and simulation uncertainty for the different flow states and different observation types

In this study, the strength of this confinement is used to assess the information content of the set of observations during the different flow states. The strength of the confinement is quantified by the reduction of the distance between the 25th and 75th percentile of each model parameter after the confinement through the soft rules. For instance, parameter $c_{SO4}$ (Table 2) has the prior range of $0 - 100$ mg·l⁻¹. Consequently, the uniform sampling strategy for the initial sample will result in values close to 25 and 75 mg·l⁻¹ for the 25th and the 75th percentile, respectively. Applying one of the soft rules may now result in values of 10 and 30 mg·l⁻¹ for the 25th and the 75th percentile, respectively. Hence, the reduction of the distance between the 25th and 75th percentile is 50-20 mg·l⁻¹, i.e. a reduction of 60% took place. In this example case we would find that the observations applied through the selected soft rule provided useful information to estimate this parameter. Applying this procedure for each of the four soft rules and the four time series defined by the flow states, we can assess how (1) the different types of observations (discharge, $NO_3^-$ and $SO_4^{-2}$) contribute to parameter identification, and (2) the focus on particular time periods and flow stages strengthens or weakens the confinement of the model parameters.

Particular attention is given to the comparison of the entire time period, including the times when the spring is influenced by the river, with the time periods when only the unsaturated zone and the saturated zone control the discharge of the spring. It is expected that this time period contains disinformative information for parameter estimation as the VarKarst does not take into account the river's influence. The reduction of the 25th and 75th percentile of the model parameters is used after applying the fourth soft rule (subsection 3.3) of the combined unsaturated and saturated flow state, and the entire time period including the period that is influenced by the river to understand the impact of the disinformative information on parameter identification. In a last step, the simulation uncertainty is quantified for the two time periods by plotting the simulations of the parameters sets that remained after the fourth soft rule was applied to the two observation time series. After including the disinformative time period, a greater simulation uncertainty is expected.



# 4   Results

## 4.1   Parameter estimation for the different flow states and different observation types

Different reductions of the initial sample are found by the different soft rules and during the different flow states (Figure 4). The reduction by discharge ($KGE_Q \geq 0.2$) varies among the different flow states but remains rather limited. The same is seen for the individual use of the hydrochemical information ($KGE_{NO3} \geq 0.2$ or $KGE_{SO4} \geq 0.2$). However, using the combination of all soft rules (all $KGE \geq 0.2$), a significant reduction of the initial sample is obtained for all flow states. This is most evident for the combined unsaturated and saturated state. The weakest reduction of the initial sample for all soft rules is found for the consideration of all stages including the disinformative time period influenced by the river.

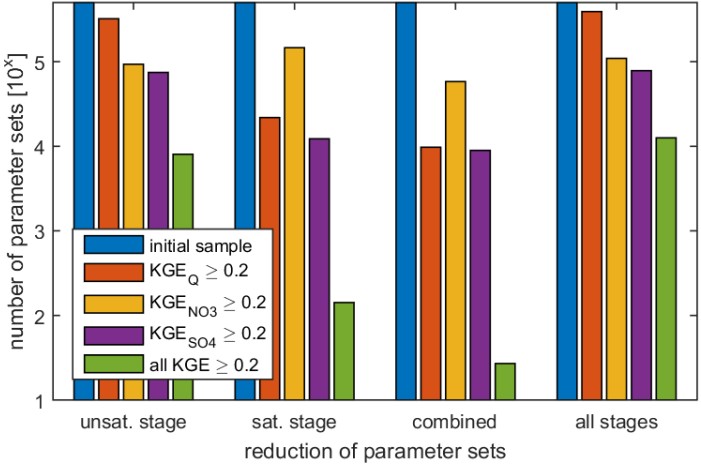

**Figure 4: Reduction of the initial sample by the four soft rules for the unsaturated state, saturated state, combined saturated and saturated states, and all system states**

The influence of the soft rules during the different flow states varies for all model parameters (Figure 5). The reduction of the initial sample by discharge ($KGE_Q \geq 0.2$) alters the uniform distribution of the initial sample for the different flow states, mostly for the parameters $A$, $V_E$ and $K_C$. These changes are most prominent in the unsaturated state ($A$), the unsaturated state ($V_E$ and $K_C$) and the combined unsaturated and saturated states ($A$, $V_E$ and $K_C$). Using $NO_3^-$ for the reduction ($KGE_{NO3} \geq 0.2$), the parameters $V_S$, $V_E$ and $a_{SE}$ experience the strongest change of their initial distribution. This change is most pronounced at the unsaturated state and the combined unsaturated and saturated states. The reduction by the observations of $SO_4^{-2}$ concentrations ($KGE_{SO4} \geq 0.2$) mostly affects the model parameters $c_{SO4}$ and $a_{SO4}$, but also find a strong impact on $a_{SE}$, mainly at the saturated state and the combined unsaturated and saturated state. Finally applying all information in the fourth soft rule (all $KGE \geq 0.2$), we find again an alteration of the model parameters that were affected by soft rules 1-3 ($A$, $V_E$, $V_S$, $a_{SE}$, $K_C$, $c_{SO4}$ and $a_{SO4}$) and, additionally, a moderate alteration of $K_E$ and $a_f$. This is most notable at the combined unsaturated





and saturated states; using all states including the disinformative period that is influenced by the river, the alterations are generally less pronounced.

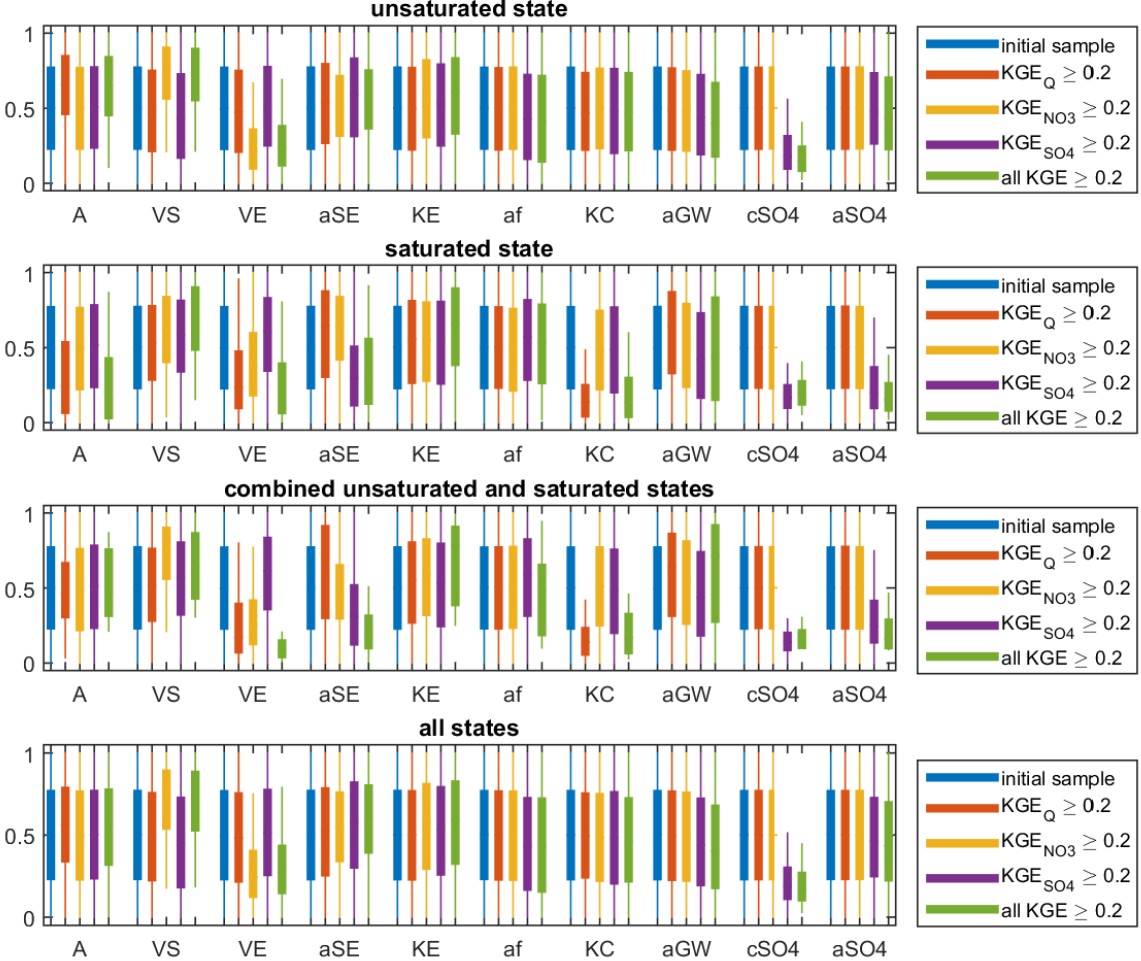

**Figure 5: Distribution of model parameters (normalised by their ranges) after applying the four soft rules for the unsaturated, saturated, and combined unsaturated and saturated, and all stages.**

## 4.2 Evaluation of information content and simulation uncertainty for the distinctive flow states and different observation types

Using the change of distance between the $25^{th}$ and $75^{th}$ percentile of each model parameter for the different soft rules and the different flow states we are able to quantify the information content of the available observations (Figure 6). We find that discharge ($KGE_Q \geq 0.2$) and $SO_4$ ($KGE_{SO4} \geq 0.2$) provide most information during the saturated flow state, while $NO_3^-$ reduces the distance between the two percentiles most during the unsaturated stage. The state that uses all information





including the disinformative time period of river influence shows generally the weakest reduction between the 25th and 75th percentile as already indicated by Figure 4.

Again the most evident changes of model parameter distributions are found for the combined unsaturated and saturated states: here, we see that observed discharge ($KGE_Q \geq 0.2$) provides most information on the parameter $K_C$, but the change of distance for the parameters $A$ and $V_E$ is still considerable. This is most evident in the combined unsaturated and saturated states. We find a more balanced distribution of information on the altered parameters when regarding the reduction obtained by $NO_3^-$ ($KGE_{NO3} \geq 0.2$). Here, the change of distances is considerable (but similar) for $V_S$, $V_E$ and $a_{SE}$. For $SO_4^{-2}$ ($KGE_{SO4} \geq 0.2$), the alteration mostly affects $c_{SO4}$, followed by a considerable alteration of $a_{SO4}$ and a moderate change of $a_{SE}$. Using all information to confine the initial sample (all $KGE \geq 0.2$) shows that the combined use of discharge, $NO_3^-$ and $SO_4^{-2}$ observations provided most information on $V_E$, $a_{SE}$, $K_C$, $c_{SO4}$, and $a_{SO4}$. Still considerable information is provide for $A$, $V_S$ and $a_f$. However, no reduction of thedistance between the 25th and 75th percentile is found for $K_E$, and even a widening takes place for $a_{GW}$.

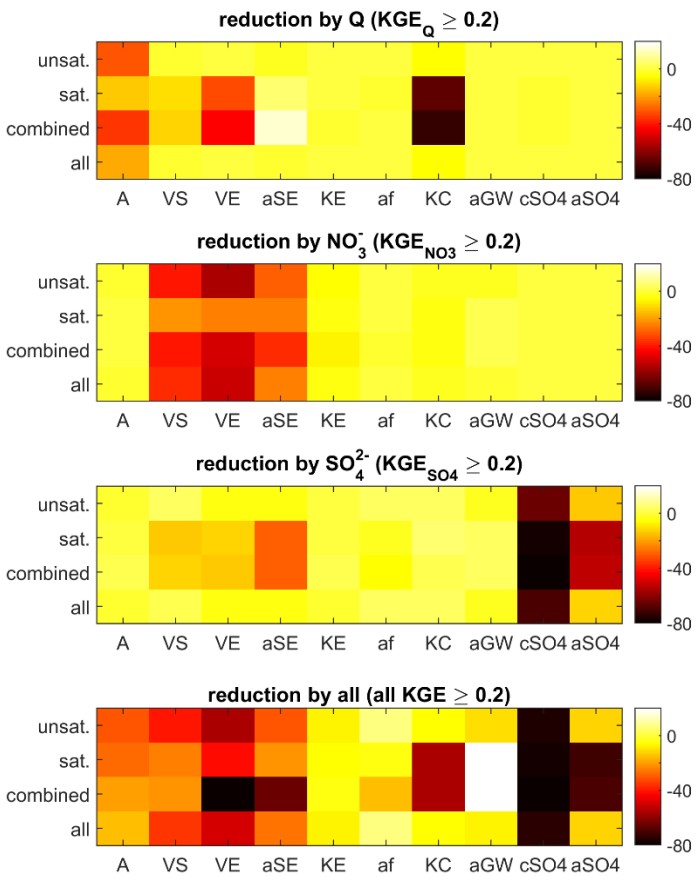

**Figure 6: Change of distance between the he 25th and 75th percentile of each model parameter when the different soft rules are applied (top to bottom) for the four flow states.**



The proceeding analysis indicates that most information to identify the largest number of model parameters is provided by the combined unsaturated and saturated flow states using discharge, $NO_3^-$ and $SO_4^{-2}$ observations. It further reveals that using the entire time period, using discharge, $NO_3^-$ and $SO_4^{-2}$ observations and including the period that is influenced by the river,

provided the fewest information; only 5 ($A$, $V_S$, $V_E$, $a_{SE}$ and $c_{SO4}$) of the 10 model parameters show a detectable reduction of the two flow percentiles (Figure 6, bottom).

The final averages of the estimated parameters (after applying 4[th] soft rule; Table 2 ) of the combined unsaturated and saturated flow states, and the state the uses the entire set of observations are similar for the parameters $A$, $V_S$ and $c_{SO4}$, while there is a strong difference for $V_E$ and $a_{SE}$. Comparing furthermore the resulting simulation uncertainty (Figure 7), we find

that the final parameters set that used all flow states, including the disinformative, river influences time period, results in a larger simulations uncertainty than the final parameter sample that used only the unsaturated and saturated flow states for parameter estimation.

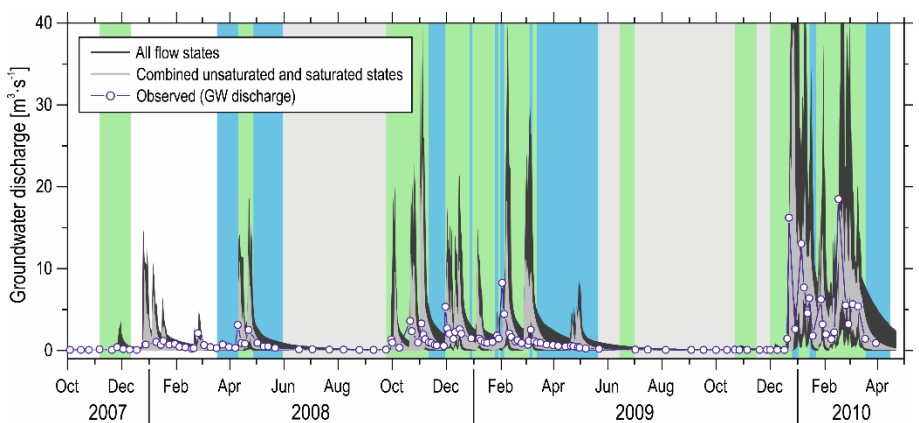

**Figure 7: Observed discharge and the simulation uncertainty of the final parameter sample (all KGE ≥ 0.2) of the combined unsaturated and saturated flow states and the all flow states including the disinformative period of river influence. Background colours representing flow states match to that of the Figure 2.**

## 5 Discussion

### 5.1 Application of the soft rules during the different flow states

The application of the 4 soft rules results in a general reduction of the initial sample for all flow states (Figure 4). A weak reduction for all of the four flow states takes place when only discharge observations are applied to confine the sample. Previous research with lumped model calibration showed that the information content of discharge observations usually suffices to calibrate 5-6 parameters (Jakeman and Hornberger, 1993; Wheater et al., 1986; Ye et al., 1997); more parameters often lead to over-parametrization (Perrin et al., 2003) and equifinality (Beven, 2006). Hence, the small reduction of the

VarKarst initial parameter sample may be due to the large number of model parameters (Table 2) with in the VarKarst





model. The same behaviour of a weak decrease of the initial parameter sample is found when the hydrochemical observations are used individually (soft rule 2 and 3). The weakest reduction of the initial parameter sample among all four flow states is found for the entire time period that includes the periods of river influence (see discussion in subsection 5.2).

When soft rule 4 (all KGE $\geq$ 0.2) is applied, we find the strongest reduction of the initial sample across all of the four flow

states. This means, the combined information of discharge, $NO_3^-$ and $SO_4^{-2}$ observations provides the most information to reduce the initial sample of model parameters. Previous research already showed that hydrochemical information can reduce parameter uncertainty (Kuczera and Mroczkowski, 1998; Rimmer and Hartmann, 2014; Son and Sivapalan, 2007). In this study, a similar reduction of parameter uncertainty could be observed (Figure 5). Depending on the applied soft rule and the considered flow states the initially uniform distributions of the model parameters are altered differently. Some model

parameters distributions change their mean without much change in the shape of their distribution (same distance between $25^{th}$ and $75^{th}$ percentile); some of the show a more confined distribution when the soft rules are applied.

## 5.2   Information provided by discharge and hydrochemistry during the different flow states

The differences of the reduction across the model parameters reveals the influence of different types of observations that were used for parameter estimation. We find that the reductions of the distance between the $25^{th}$ and $75^{th}$ percentile is most

pronounced during the saturated stage for the discharge observations (Figure 6). This indicates that discharge provides most information during the recession period. Information about hydrodynamic parameters $A$, $V_E$ and $K_C$ is derived directly from the discharge observations. This makes sense because hydrodynamic changes in the main discharge area of Sierra Blanquilla aquifer reflect the hydraulic pressure transference from the unsaturated zone to the saturated zone of the system. Similar results were found by Wagener et al. (2003) when they applied Dynamic Identifiability Analysis to a lumped rainfall runoff

model using only discharge data.

They also found that the parameters, which control the unsaturated zone and fast flow components of their model, are most identifiable during and just after the rainfall-runoff events. Our results indicates a similar behaviour by showing the strongest reduction of the distance between the $25^{th}$ and $75^{th}$ percentile for the unsaturated zone parameters during the unsaturated flow state using the $NO_3^-$ observations (parameters $V_S$, $V_E$ and $a_{SE}$). This is in accordance with Reusser and Zehe (2011) who

showed that model parameters that control the recession period are most sensitive during the recession period with a time dynamic resection and cluster analysis using discharge information. $NO_3^-$ has been used almost as an ideal tracer to determine infiltration processes through the soil and epikarst in the shallower aquifer zones (Hunkeler and Mudry, 2007; Mudarra et al., 2014). Thus, $NO_3^-$ observations contribute stronger to the identification of surface and evapotranspiration processes during the unsaturated flow state. This can be explained by the relative stability of $NO_3^-$ dynamics within the karst

system under oxidizing conditions (Mudarra et al., 2014), which favours its preservation from surface to the spring.

$SO_4^{-2}$ provided most information on the parameters $c_{SO4}$, $a_{SO4}$, and $a_{SE}$ during the saturated state. This makes sense as $SO_4^{-2}$ is stored within saturated zone of the system where groundwater is in touch with gypsum-bearing geological formations



(Triassic clays with evaporites), which are found in contact with deeper aquifer compartments. $SO_4^{-2}$ time series provide more information about the unsaturated zone / epikarst drainage during the saturated flow stage. Such findings mean that the high chemical contrast observed in $SO_4^{-2}$ concentrations of fresh (recently infiltrated) and old (stored) groundwater is useful to assess the relative importance of unsaturated flow and saturated flow during saturated flow stage (Barberá and Andreo, 2015; Mudarra et al., 2011).

The highest number of identifiable parameters is found when all information (discharge, $NO_3^-$ and $SO_4^{-2}$ observations) are combined and estimated during the combined unsaturated and saturated flow stages (Figure 6). In addition to the parameters that showed an increase of identifiability at the individual stages ($A$, $V_E$, $K_C$, $V_S$, $V_E$ , $a_{SE}$, $c_{SO4}$, and $a_{SO4}$), we also see an increased identifiability of the parameter $a_f$, most probably due to parameter interactions (Pianosi et al., 2016). Compared to that, using all information during the entire time period, including the disinformative period, only five of the model parameters show a visible decrease of the distance between the 25[th] and 75[th] percentile of their distribution. Hence, the inclusion of the disinformative period led to an increase of posterior parameter uncertainty compared to using only the informative time periods represented by the unsaturated and saturated states. This was also shown by Beven and Westerberg (2011) or Beven et al. (2011), when they considered the impact disinformative discharge events.

The impact of the disinformative time period on the precision of the observations is clearly visible in Figure 7. Since the model has to compensate for structural errors, i.e. the missing representation of the influence of the river on the discharge of the karst spring, it is forced to allow for a wider range of parameter combinations to account for the simulation errors. Using only the unsaturated and saturated states allows for a much better confinement of model parameters and therefore a much smaller simulation uncertainty, although showing some deviations during the periods when the fiver affects the flow system of the spring (blue shaded areas in Figure 7). Hence, similar to Kauffeldt et al. (2013), our study shows that a proper pre-analysis of the information content observations for model parameters estimation (subsection 2.2) allows for excluding disinformative information to reduce model parameter and simulation uncertainty.

## 5.3    Limits and transferability of the approach

The analysis of variations of the groundwater component in the Turón River flow has permitted to determine the timing, duration and magnitude of the global hydrodynamic aquifer responses under influenced hydrological conditions, as well as to assess the discharge thresholds from which different compartments (i.e. flooding of relict conduit networks) of the behaviour (Barberá and Andreo, 2015). However, a more accurate decomposition of flow components from the study of spring hydrographs has not been possible due to the relatively low resolution of discharge time series (Table 1). Even though the chemical signature of groundwater that drains the different aquifer zones (unsaturated zone and saturated zone), and that is affected by the Turón river, can be estimated using electric conductivity (Figure 2), it is rather based on subjective interpretation. However, it can be argued that the previous knowledge based on the accurate interpretation of El Burgo spring chemographs has permitted a realistic flow decomposition from EC time series as our results show a clear difference of



estimated parameter distributions and resulting simulation uncertainty using the unsaturated and saturated flow states and the entire time period including the disinformative data. A more precise distinction between the states is only possible if specific chemical indicators are available better constrain the differentiation of flow states contributing to El Burgo spring discharge, which was not possible within the frame of this study. But even though due to subjectivity, the identification of time periods

or data sets that disinformative contributions to parameter estimation is a useful way to reduce the simulation uncertainty of hydrological models. Building on previous research on disinformative data that focussed on disinformative discharge information, our approach provides a systematic procedure that also includes hydrochemical observations to identify disinformative periods and to improve parameter estimation of models for complex hydrological systems.

## 6    Conclusions

In this research, a new approach to estimate the information content of water quality data and the value of identifying most informative periods for model parameter estimation has been proposed. Using soft rules to include discharge, $NO_3^-$ and $SO_4^{-2}$ observations into the parameter estimation procedure, we were able to reduce an initial sample of 500,000 parameter sets during pre-defined flow states; one of the including a known period of disinformative data. Comparing the distributions of the initial and reduced parameter sets, we were able to quantify the information contained in our observations to identify the

parameters of our simulation model.

We found that the information content of the observations varies for the different states that we considered. $NO_3^-$ provided most of its information when the unsaturated zone processes dominate the discharge behaviour of the spring. During the time when the saturated zone controls the outflow behaviour, $SO_4^{-2}$ and discharge observations provide the best information to identify the model parameters. Including the disinformative period, the information content of all data generally decreases,

as well as the uncertainty of simulations increases. We finally show that the combination of saturated and unsaturated flow states provides the most precise information about the model parameters. Due to parameter interactions, even model parameters that were not identifiable during the unsaturated or saturated flow state alone became identifiable. As a result, the simulation uncertainty is significantly reduced compared to the simulations obtained by the entire time series of observations that include the disinformative data.

Even though exemplified at a particular karst spring in Southern Spain, our approach is easily transferrable to other modelling studies that want to use water quality data for the identification of disinformative periods and for the estimation of model parameters. Our results add to previous findings on the value of removing disinformative data from model parameter estimation to reduce simulation uncertainty. Furthermore our results can help building a better communication between experimental hydrologists and modellers (Seibert and McDonnell, 2002) as hydrochemical data is often used for system

characterization. Our study showed that $NO_3^-$ and $SO_4^{-2}$ that often are used for understanding the unsaturated and saturated zone processes also help to identify the corresponding process parameters in our model. Further research should therefore include the evaluation of other hydrochemical variables that can be attributed to particular hydrological processes to and their value to identify the corresponding processes in process-based simulation models.



## 7    Acknowledgements

This work is a contribution to the projects P06-RNM 2161 of Junta de Andalucía; and CGL2008-06158 BTE, CGL2012-32590, CGL2015-65858-R of DGICYT and to the Research Group RNM-308 of the Junta de Andalucía. The article processing charge was funded by the German Research Foundation (DFG) and the University of Freiburg in the funding
programme Open Access Publishing.

## 8    Appendix

The parameter $V_{mean,S}$ [mm] and the distribution coefficient $a_{SE}$ [-] control the variability of soil depths over the $N$ model compartments. Using them, the soil storage capacity $V_{S,i}$ [mm] for every compartment $i$ is defined by:

$$V_{S,i} = V_{\max,S} \cdot \left( \frac{i}{N} \right)^{a_{SE}} \tag{3}$$

$V_{max,S}$ [mm] represents the maximum soil storage capacity and is derived from $V_S$:

$$\int_0^{i_{1/2}} V_{\max,S} \left( \frac{x}{N} \right)^{a_{SE}} dx = \frac{\int_0^N V_{\max,S} \left( \frac{x}{N} \right)^{a_{SE}} dx}{2} ; V_S = V_{\max,S} \left( \frac{i_{1/2}}{N} \right)^{a_{SE}}$$

$$\Updownarrow$$

$$V_{\max,S} = V_S \cdot 2^{\left( \frac{a_{SE}}{a_{SE}+1} \right)} \tag{4}$$

Where the compartment at which the volumes on the left equal the volumes on the right is found at $i_{1/2}$. The same distribution coefficient $a_{SE}$ is used to derive the epikarst storage distribution by the mean epikarst depth $V_E$ [mm] (derivation of $V_{max,E}$ likewise to $V_{max,S}$ in Eq (4)):

$$V_{E,i} = V_{\max,E} \cdot \left( \frac{i}{N} \right)^{a_{SE}} \tag{5}$$

Actual evapotranspiration from each soil compartment $E_{act,i}$ is calculated by:

$$E_{act,i}(t) = E_{pot}(t) \cdot \frac{\min\left[V_{Soil,i}(t) + P(t) + Q_{Surface,i}(t), V_{S,i}\right]}{V_{S,i}} \tag{6}$$

Potential evapotranspiration $E_{pot}$ [mm] is found by the Thornthwaite equation (Thornthwaite, 1948) and $Q_{surface,i}$ [mm] is the surface inflow that originates from compartment $i$-1 (see Eq. (11)). $V_{Soil,i}$ [mm] is the volume of water stored in the soil at
time step $t$. Recharge from the soil to the epikarst $R_{Epi,i}$ [mm] is found by water balance:

$$R_{Epi,i}(t) = Q_{\inf}(t) + \max\left[V_{Soil,i}(t) + P(t) + Q_{Surface,i}(t) - E_{act,i}(t) - V_{S,i}, 0\right] \tag{7}$$



with $Q_{inf}(t)$ being the river infiltration (Eq. (5)). The epikarst storage coefficients $K_{E,i}$ [d] controls the outflow from the epikarst:

$$Q_{Epi,i}(t) = \frac{\min\left[V_{Epi,i}(t) + R_{Epi,i}(t), V_{E,i}\right]}{K_{E,i}} \cdot \Delta t \tag{8}$$

$$K_{E,i} = K_{max,E} \cdot \left(\frac{N-i+1}{N}\right)^{a_{SE}} \tag{9}$$

Here, $V_{Epi,i}$ [mm] is the water stored in the epikarst at time step $t$. $K_{max,E}$ is found by the mean epikarst storage coefficient $K_E$ and by applying the same distribution coefficient $a_{SE}$:

$$N \cdot K_E = \int_0^N K_{max,E} \left(\frac{x}{N}\right)^{a_{SE}} dx$$
$$\Updownarrow \tag{10}$$
$$K_{max,E} = K_E \cdot (a_{SE} + 1)$$

Surface flow to the next model compartment $Q_{Surf,i+1}$ [mm] initiates when soil and epikarst storage capacities are exceeded:

$$Q_{Surf,i+1}(t) = \max\left[V_{Epi,i}(t) + R_{Epi,i}(t) - V_{E,i}, 0\right] \tag{11}$$

The vertical percolation from the epikarst is split into diffuse ($R_{diff,i}$ [mm]) and concentrated groundwater recharge ($R_{conc,i}$ [mm]) again by a variable separation factor $f_{C,i}$ [-] and a distribution coefficient $a_f$ [-]:

$$R_{conc,i}(t) = f_{C,i} \cdot Q_{Epi,i}(t) \tag{12}$$

$$R_{diff,i}(t) = (1 - f_{C,i}) \cdot Q_{Epi,i}(t) \tag{13}$$

$$f_{C,i} = \left(\frac{i}{N}\right)^{a_f} \tag{14}$$

The diffuse recharge reaches the groundwater compartments ($i = 1 \ldots N$-1) directly below, while concentrated recharge is routed laterally to the conduit system (compartment $i = N$, :). Similar to epikarst storage coefficients, variable groundwater storage coefficients $K_{GW,i}$ [d] are calculated. The, groundwater contributions of the matrix system $Q_{GW,i}$ [mm] in therefore found by:

$$Q_{GW,i}(t) = \frac{V_{GW,i}(t) + R_{diff,i}(t)}{K_{GW,i}}; i = 1 \ldots N-1 \tag{15}$$

with

$$K_{GW,i} = K_C \cdot \left(\frac{i}{N}\right)^{-a_{GW}} \tag{16}$$



The conduit system discharges from compartment $N$:

$$Q_{GW,i}(t) = \frac{V_{GW,N}(t) + \sum_{i=1}^{N} R_{conc,i}(t)}{K_C}; i = N \tag{17}$$

where conduit storage coefficient is given by $K_C$ [d]. The discharge of the main spring $Q_{main}$ [l s$^{-1}$] is comprised by the sum of the matrix and the conduit system discharge rescaled to [l s$^{-1}$] the recharge area $A$ [km²]:

$$Q_{main}(t) = \frac{A}{N} \cdot \sum_{i=1}^{N} Q_{GW,i}(t) \tag{18}$$

Solute transport within the VarKarst model follows the assumption of complete mixing for every model compartment. Hence, enrichment only takes place due to evaporation and by geogene dissolution (only SO$_4^{2-}$), for which varying equilibrium concentrations are defined according to:

$$c_{SO4,i} = c_{max,SO4} \cdot \left(\frac{N - i + 1}{N}\right)^{a_{SO4}} \tag{19}$$

where $a_{SO4}$ is a variability constant and $c_{max,SO4}$ is derived from $c_{SO4}$ [mg l$^{-1}$] (similar to eq. (10)).

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
