# Peer review of "On the value of water quality data and informative flow states in karst modelling"

_Hydrology and Earth System Sciences, 2017_

## Referee Comment (RC1) · 24 Jun 2017

This study examines the information content of water quality data for a karst simulation model. The Varkarst model is applied to a spanish watershed. An initial 500.000 random parameters sets is confined using discharge and water quality data which are either taken separately or together, using the whole time series of focusing on specific flow stages. The reduction in the 25th to 75th percentiles range is used to estimate the information content of the data.

The methodology is clear, the results are commented with adequate references to related works, the illustrations are sufficient and informative. In my opinion, the manuscript deserves to be published after minor revisions.

[Figure]

General comments

- The model equations are detailed in annex but for more clarity some information about the solute model should appear in Section 3. The lack of NO3-specific parameter is surprising and should be commented.

- Parameter kE seems unsensitive. Is this related to some specificity of the flow processes on the test site ?

- The KGE is nicely defined as a combination of linear correlation and the ratios of the mean and standard deviations of the simulations and observations. Have you had a look at whether the parameters have more influence on r, alpha or beta ?

Technical comments

- p 5 l 14 "wtaer"

- p 7 l 16 "the fore" instead of "therefore"

- p 10 l 14 "the unsaturated state (VE Kc)" shoud be corrected in ""the saturated state (VE Kc)"

- p12 l 10 "is provide"

- p 12 caption of Figure 6: "the he 25th"

- p 15 l 26 something is missing in the sentence "discharge thresholds from wich different compartments (...) of the behaviour"

---

## Referee Comment (RC2) · A. Malard (Referee) · 13 Jul 2017

This paper intends to provide an approach for reducing uncertainties in the Varkarst simulation model (= lumped model divided in compartments). 500'000 parameters sets have been confined using discharge, $NO_3^-$ and $SO_4^{2-}$ measurements, (i) together or in a separate way and (i) applied on the whole time series or in sub-series corresponding to expected flow processes (floods, recession, mid-stages). Besides, datasets have been resampled in the range of the 25th to 75th percentiles using soft rules in order to assess how the observations contribute to describe the parameter. Finally, repeated simulations using the reduced 250'000 sets of parameters make it possible for the authors to identify that: - "$NO_3^-$ provides most information to identify the model parameters controlling soil and epikarst dynamics for unsaturated -flow state (i.e. flood

events" - "SO42- and discharge data provides most information to identify the model parameters for saturated-flow state (i.e. recession periods).

The approach sounds coherent but authors might provide more information on the model timestep and the timestep used for applying the Kling-Gupta coefficient. Indeed, measurements are of lower resolution and it is not mentioned how the authors managed that.

A few other comments - and minor corrections in the attached .pdf

Few more words on the soft rules would also be appreciated

Please also note the supplement to this comment:
https://www.hydrol-earth-syst-sci-discuss.net/hess-2017-230/hess-2017-230-RC2-supplement.pdf

———————————————

---

## Author Comment (AC2) · 13 Aug 2017

**Reviewer #1 (Arnauld Malard)**

*This paper intends to provide an approach for reducing uncertainties in the Varkarst simulation model (= lumped model divided in compartments). 500'000 parameters sets have been confined using discharge, NO3- and SO42- measurements, (i) together or in a separate way and (i) applied on the whole time series or in sub-series corresponding to expected flow processes (floods, recession, mid-stages). Besides, datasets have been resampled in the range of the 25th to 75th percentiles using soft rules in order to assess how the observations contribute to describe the parameter. Finally, repeated simulations using the reduced 250'000 sets of parameters make it possible for the authors to identify that: - "NO3- provides most information to identify the model parameters controlling soil and epikarst dynamics for unsaturated -flow state (i.e. flood events" - "SO42- and discharge data provides most information to identify the model parameters for saturated-flow state (i.e. recession periods).*

*The approach sounds coherent but authors might provide more information on the model timestep and the timestep used for applying the Kling-Gupta coefficient. Indeed, measurements are of lower resolution and it is not mentioned how the authors managed that.*

*A few other comments - and minor corrections in the attached .pdf*

*Few more words on the soft rules would also be appreciated*

**Our response:** We thank Dr Malard for his valuable recommendations. In the revised manuscript, we will provide a more detailed model description (as also recommended by Dr Mazzilli in her review) including more information about the temporal resolution of the model and the observations and how they were linked within the parameter estimation. Also, some more elaboration on the soft rules will be provided in the methods section. Both issues, the impact of lower resolution of the measurements, as well as the impact of variations in the soft rules will be discussed in more detail in the discussion section of the revised manuscript.

**Specific and technical comments from commented pdf**

*P5L3: For some event spring's peaks discharge seem to be comcomitant with EC depletion...*

*The resolution of the flow measurements (1 measure/week) reveals insufficient to ensure the supposed concomittancy...*

**Our response:** True, we will rephrase this statement and add some discussion on the uncertainties that go along with such resolution discrepancies.

*P5L17: From where? Epikarst, Unsaturated zones or drainage of the phreatic zone?*

**Our response:** The phreatic zone. We will clarify this in the revised manuscript.

*P5L18: "Seepage" from the epikarst should not be disregarded...*

**Our response:** We agree that seepage from the epikarst will still be abundant during this stage. We will clarify this and add some elaboration why we believe that our distinction of flow states still makes sense.

*P6L10: daily timescale?*

**Our response:** Yes, daily time scale. We will provide this information in the revised version of the paper.

*P8L15: How did you manage the differences in time step between model (daily) vs. measurements (biweekli)?*

**Our response:** Simulations and observations are only compared by KGE at times when observations are available. If the resolution of observations were higher, more parameter sets could have been discarded by our soft rules and the precision of the simulation with the remaining parameter sets would have been higher. We will add this important information to the methods and discussion section.

*P9L26: The "combined" state should be explicitly mentioned in Figure 1.*

**Our response:** We will update Figure 1 accordingly.

*P10L14: replace: "saturated"*

**Our response:** Yes, the word will be replaced.

*P11: marked areas in Fig 5*

**Our response:** Unfortunately, there is no comment explaining the marked areas in Fig 5. For the revisions, we will assume that they were only included to facilitate the review but do not require any modification to the manuscript.

*P11L10: Use "SO42-"*

**Our response:** We will correct all hydrochemical variables accordingly.

*P12L10: typo*

**Our response:** Typo will be corrected.

*P12L10: typo*

**Our response:** Typo will be corrected.

*P13L8: clarify*

**Our response:** We refer to the entire time period that also includes the periods of river influence. The statement will be clarified accordingly.

*P13: Please Make this figure bigger...(Fig 7)*

**Our response:** The figure will be enlarged in the new version of the manuscript.

*P13L25: typo*

**Our response:** Typo will be corrected.

*P16L13: ...; one being a known period...*

**Our response:** The sentence will be changed accordingly.

---

## Author Response (AR1)

Freiburg, October 11th, 2017

Dear editor,

we are happy to provide the revised version of our manuscript

**On the value of water quality data and informative flow states in karst modelling**.

According to the remarks of the two referees, Dr Naomi Mazzilli and Dr Arnauld Malard, we added more information on the solute transport approach of the model, the Kling-Gupta efficiency and the problem of the varying temporal resolution of the discharge end hydrochemical measurements.

Please find below a detailed point-by-point response to the reviewer comments and a changes-tracked version of the manuscript.

We want to thank the associate editor and the two referees for their valuable remarks that led to a significant improvement of our original submission and we hope that the manuscript can now be considered for publication in Hydrology and Earth System Sciences.

Sincerely,
* * *
Andreas Hartmann
Juan Antonio Barberá
Bartolomé Andreo

**Reviewer #1 (Naomi Mazzilli)**

*This study examines the information content of water quality data for a karst simulation model. The Varkarst model is applied to a spanish watershed. An initial 500.000 random parameters sets is confined using discharge and water quality data which are either taken separately or together, using the whole time series of focusing on specific flow stages. The reduction in the 25th to 75th percentiles range is used to estimate the information content of the data.*

*The methodology is clear, the results are commented with adequate references to related works, the illustrations are sufficient and informative. In my opinion, the manuscript deserves to be published after minor revisions.*

**Our response:** We thank Dr Mazzilli for her positive and valuable review.

**General comments**

*• The model equations are detailed in annex but for more clarity some information about the solute model should appear in Section 3. The lack of NO3-specific parameter is surprising and should be commented.*

**Our response:** The model description and an elaboration of the $NO_3^-$ dynamics was be improved in the revised version of the manuscript (subsection 3.2):

> "Solute transport simulations within the model follow the assumption of instantaneous and complete mixing within each storage (soil, epikarst, groundwater) and each of the N model compartments (Figure 3). In the particular case of $NO_3^-$, this implies neglecting plant uptake and release processes, which were found to be important in more humid regions (Hartmann et al., 2016) but it was found a valid assumption at Mediterranean regions such as our study site (Hartmann et al., 2013b, 2014b)."

*• Parameter kE seems unsensitive. Is this related to some specificity of the flow processes on the test site ?*

**Our response:** The parameter KE often interacts with the parameter VE. A discussion of the consequences of this interaction was added to the revised manuscript (Subsection 5.2):

> "Only $a_{GW}$ and KE remain with low identifiability, which may be due to structural limitations of the model structure (Clark et al., 2008) or due to parameter interactions that are not explicitly considered in our approach. In fact, a lower identifiability of $K_E$ in favour of a high identifiability of VE was found in a previous study with a similar version of the model (Hartmann et al., 2015).

*• The KGE is nicely defined as a combination of linear correlation and the ratios of the mean and standard deviations of the simulations and observations. Have you had a look at whether the parameters have more influence on r, alpha or beta ?*

**Our response:** This is a very interesting remark. An analysis or the influence of r, alpha and beta was partially done in Hartmann *et al.* (2013). In the case of this study, we omitted the analysis of the individual components of the KGE as we already considered three different variables (discharge, $NO_3^-$ and $SO_4^{2-}$) and we did not want to confuse the reader by adding more dimensions to this evaluation (also we believe that the results we get is sufficient to make our point). However, we agree that now having quantified the information content of the different data types, a follow up study should

analyse in more detail how the model performs for r, alpha and beta individually. We added a respective statement to the conclusions:

> "Also, a further disaggregation of the Kling Gupta efficiency in its components, correlation, bias and variability, contains high promise for further advance of our approach."

**Technical comments**

• p 5 l 14 "wtaer"

• p 7 l 16 "the fore" instead of "therefore"

• p 10 l 14 "the unsaturated state (VE Kc)" shoud be corrected in ""the saturated

state (VE Kc)"

• p12 l 10 "is provide"

• p 12 caption of Figure 6: "the he 25th"

• p 15 l 26 something is missing in the sentence "discharge thresholds from wich different compartments (...) of the behaviour"

**Our response:** Corrections were applied to address all technical comments.

**Reviewer #2 (Arnauld Malard)**

*This paper intends to provide an approach for reducing uncertainties in the Varkarst simulation model (= lumped model divided in compartments). 500'000 parameters sets have been confined using discharge, NO3- and SO42- measurements, (i) together or in a separate way and (i) applied on the whole time series or in sub-series corresponding to expected flow processes (floods, recession, mid-stages). Besides, datasets have been resampled in the range of the 25th to 75th percentiles using soft rules in order to assess how the observations contribute to describe the parameter. Finally, repeated simulations using the reduced 250'000 sets of parameters make it possible for the authors to identify that: - "NO3- provides most information to identify the model parameters controlling soil and epikarst dynamics for unsaturated -flow state (i.e. flood events" - "SO42- and discharge data provides most information to identify the model parameters for saturated-flow state (i.e. recession periods).*

*The approach sounds coherent but authors might provide more information on the model timestep and the timestep used for applying the Kling-Gupta coefficient. Indeed, measurements are of lower resolution and it is not mentioned how the authors managed that.*

*A few other comments - and minor corrections in the attached .pdf*

*Few more words on the soft rules would also be appreciated*

**Our response:** We thank Dr Malard for his valuable recommendations. In the revised manuscript, we provided a more detailed model description (as also recommended by Dr Mazzilli in her review) including more information about the temporal resolution of the model and the observations and how they were linked within the parameter estimation (subsection 3.3):

> "For the calculation of KGE, only time steps, at which observations are available, are considered. Hence, the KGE values will only express the model performance to reflect the discharge, $NO_3^-$ and $SO_4^{2-}$ observations that were sampled in a 7-8 days temporal resolution (Table 1) even though the model runs on a daily time step."

Also, some more elaboration on the soft rules was provided in the methods section (subsection 3.3):

> "The threshold value of 0.2 was found by preliminary analysis. Its rather low value is meant to take into account that the simulation is exposed to various sources of uncertainty including uncertainties of the model input (observation of climate variables and their application to the entire recharge area), model structure uncertainty (representation of karst processes by conceptual mathematical formulations in a semi-distributed way), and the uncertainty of observations (discharge measurement and hydrochemical analysis, as well as their low temporal resolution)."

Both issues, the impact of lower resolution of the measurements, as well as the impact of variations in the soft rules is now discussed in more detail in the discussion section of the revised manuscript (subsection 5.3):

> "Another limitation of our research is the low resolution of the discharge and hydrochemical observations (7-8 days). Although our approach took into account this weakness by the soft rules allowing for remaining uncertainty after the reduction of our 500,000 parameter sets, we believe that a higher resolution of the observations (preferably 1 day) would have resulted in a more pronounced reduction of the initial sample and consequently to a lower remaining uncertainty."

**Specific and technical comments from commented pdf**

*P5L3: For some event spring's peaks discharge seem to be comcomitant with EC depletion...*

*The resolution of the flow measurements (1 measure/week) reveals insufficient to ensure the supposed concomittancy...*

**Our response:** True, we rephrased this statement.

*P5L17: From where? Epikarst, Unsaturated zones or drainage of the phreatic zone?*

**Our response:** The phreatic zone. We clarified this in the revised manuscript.

*P5L18: "Seepage" from the epikarst should not be disregarded...*

**Our response:** We agree that seepage from the epikarst will still be abundant during this stage. We clarified this and added some elaboration why we believe that our distinction of flow states still makes sense (subsection 2.2):

> "Even though, there still might be some seepage from the soil and epikarst during this stage, the hydrochemical signature of the spring, which is dominated by the signal of the phreatic zone (Barberá and Andreo, 2015), shows that these fractions are not very important."

*P6L10: daily timescale?*

**Our response:** Yes, daily time scale. We provided this information in the revised version of the paper.

*P8L15: How did you manage the differences in time step between model (daily) vs. measurements (biweekli)?*

**Our response:** Simulations and observations are only compared by KGE at times when observations are available. If the resolution of observations were higher, more parameter sets could have been discarded by our soft rules and the precision of the simulation with the remaining parameter sets

would have been higher. We added this important information to the methods and discussion section (please see our response to the general comment of this review).

*P9L26: The "combined" state should be explicitly mentioned in Figure 2.*

**Our response:** We updated Figure 2 and its caption accordingly.

*P10L14: replace: "saturated"*

**Our response:** The word was be replaced.

*P11: marked areas in Fig 5*

**Our response:** Unfortunately, there is no comment explaining the marked areas in Fig 5. For the revisions, we will assume that they were only included to facilitate the review but do not require any modification to the manuscript.

*P11L10: Use "SO42-"*

**Our response:** Corrected.

*P12L10: typo*

**Our response:** Corrected.

*P12L10: typo*

**Our response:** Corrected.

*P13L8: clarify*

**Our response:** We refer to the entire time period that also includes the periods of river influence. The statement was clarified accordingly.

*P13: Please Make this figure bigger...(Fig 7)*

**Our response:** The figure was enlarged in the new version of the manuscript.

*P13L25: typo*

**Our response:** Corrected.

*P16L13: ...; one being a known period...*

**Our response:** The sentence was changed accordingly.

[revised manuscript text omitted]